# Differences in Physiological Performance and Gut Microbiota between Deep-Sea and Coastal Aquaculture of Thachinotus Ovatus: A Metagenomic Approach

**DOI:** 10.3390/ani13213365

**Published:** 2023-10-30

**Authors:** Shuangfei Li, Shilin Wang, Cong Pan, Yanqing Luo, Shitong Liang, Siru Long, Xuewei Yang, Boyu Wang

**Affiliations:** 1Guangdong Technology Research Center for Marine Algal Bioengineering, Guangdong Key Laboratory of Plant Epigenetics, College of Life Sciences and Oceanography, Shenzhen University, Shenzhen 518060, China; sfli@szu.edu.cn (S.L.); linseyw0521@163.com (S.W.); cp18683725070@163.com (C.P.); lyanqingl87@163.com (Y.L.); tong_09090614@163.com (S.L.); srlong00@163.com (S.L.); 2Shenzhen Key Laboratory of Marine Biological Resources and Ecology Environment, College of Life Sciences and Oceanography, Shenzhen University, Shenzhen 518055, China; 3Longhua Innovation Institute for Biotechnology, Shenzhen University, Shenzhen 518060, China

**Keywords:** mariculture, deep sea, coastal, growth performance, intestinal microbiota, metagenome sequence

## Abstract

**Simple Summary:**

In this manuscript, we determined the growth performance and physiological and biochemical indices of the aquaculture fish *Trachinotus ovatus* reared in deep-sea and coastal environments, and analyzed the species abundance and diversity of the fish’s intestinal microbes using high-throughput sequencing to screen for antibiotic resistance genes. We also analyzed environmental water samples to identify possible reasons for the observed differences between the two environments. The study found no significant difference in growth performance between deep-sea cultured fish and traditional coastal cultured fish during the feeding cycle. However, the physiological and biochemical indices and the number of antibiotic resistance genes in the gut microbial community of deep-sea cultured fish were superior to those of their coastal counterparts. Coastal cultured fish are particularly vulnerable to the impact of domestic water discharged from human activities on shore. This may be the cause of the observed phenomenon. These studies can offer additional data for comparing deep-sea aquaculture with traditional coastal aquaculture and provide a reference for the future development of fishery transformation and scientific standardization of deep-sea aquaculture.

**Abstract:**

Aquaculture has become the fastest growing sector in global agriculture. The environmental degradation, diseases, and high density of mariculture has made for an inevitable shift in mariculture production from coastal to deep-sea areas. The influence that traditional coastal and emerging deep-sea farming environments exert on aquatic growth, immunity and gut microbial flora is unclear. To address this question, we compared the growth performance, physiological indicators and intestinal microbiological differences of deep-sea and coastal aquaculture in the Guangxi Beibu Gulf of China. The results showed that the growth performance and the complement of C3 and C4 (C3, C4), superoxide dismutase (SOD), and lysozyme (LYS), these physiological and biochemical indicators in the liver, kidney, and muscle of *Trachinotus ovatus* (*T. ovatus*), showed significant differences under different rearing conditions. Metagenome sequencing analysis showed *Ascomycota*, *Pseudomonadota*, and *Bacillota* were the three dominant phyla, accounting for 52.98/53.32 (coastal/deep sea), 24.30/22.13, and 10.39/11.82%, respectively. Aligned against the CARD database, a total of 23/2 (coastal/deep-sea) antibiotic resistance genes were screened and grouped into 4/2 genotypes. It indicated that compared with deep-sea fish, higher biological oxygen levels (3.10 times), inorganic nitrogen (110.00 times) and labile phosphate levels (29.00 times) in coastal waters might contributed to the existence of eutrophication with antibiotic resistance. The results of the study can provide complementary data on the study of the difference between deep-sea farming and traditional coastal farming, serving as a reference to future in-depth work on the transformation of fisheries development and scientific standardization of deep-sea farming.

## 1. Introduction

The rapid expansion in demand for marine resources and in ecological contamination have resulted in the depletion and decline of aquatic resources, which has become a severe global issue [1]. The large-scale, high-density coastal mariculture, coastal human activities, and land-based discharges are causing coastal seawater eutrophication and other environmental concerns [2]. In this regard, an increasing number of countries are actively investigating ecologically friendly methods of agriculture and pursuing deep-sea mariculture development [3]. The migration of aquaculture activities from safe coastal locations to more exposed areas with heavy wave action and strong ocean currents is known as deep-sea or pelagic aquaculture [4,5]. Generally, nearshore marine aquaculture is limited by ecological carrying capacity due to lack of available area, eutrophication from effluent discharge, and benthic ecosystems, while deep-sea aquaculture is less affected by socio-economic tensions and pollution [6,7].

The current deep-sea aquaculture approach primarily focuses on optimizing aquaculture technology on a macroscale, exploring the interplay between deep-sea aquaculture and environmental quality and the influence of deep-sea aquaculture on biological populations [8]. For example, it has been proposed to investigate the impact of human activities, including deep-sea farming, on marine ecology by the Decade of Ocean Science for Sustainable Development [9]. The excessively high stocking density impedes fish farming development in deep-sea nets and lowers feed-utilization efficiency [10]. Moreover, the aforementioned issues, the comparison of differences between deep-sea and coastal culture, such as changes in microbial communities [11] and the influence of human activities on culture [12], have received increasing interest recently [13]. For instance, Pogoda found that oysters cultivated in deep-sea areas grew faster than oysters produced on land [5]. Furthermore, deep-sea fish can be utilized to obtain safe mineral sources for the human diet [4].

Microbiota in the intestine may play a role in avoiding pathogen colonization and promoting beneficial activities, such as inducing innate immune responses in the host [14]. For instance, probiotics can adhere to the intestinal mucosa and facilitate nutrient exchange [15]. Similarly, the gut microbiota promotes the ability of fish to absorb lipids from the diet [16], which causes them to accumulate fat more efficiently [17]. *T. ovatus* is an omnivorous hard-boned marine fish with tender meat and a delicious taste [18], and it is also a valuable edible fish in the whole world [19]. The current study of *T. ovatus* is focused mostly on feed nutrition ratio, growth cycle and artificial breeding technologies. For example, Zhao discovered that alterations in the gut microbiota of *T. ovatus* can be caused by starch levels in feed [20]. And there are reports about how high-density farming affects the growth of *T. ovatus* [10]. Since the gut microbiota plays a crucial role in shaping fish farming, fewer data are available on the gut microbiota of the important economic fish *T. ovatus*. Furthermore, exploring the differential influence of the growth and gut microbiota between coastal and deep-sea fish will open ways to artificial breeding technologies.

In this study, the growth performance of *T. ovatus* cultured in two different areas, deep-sea and coastal, was investigated. The physiological and biochemical indicators from analysis of liver, kidney, and muscle tissues of *T. ovatus* were evaluated. Metagenomic analysis was used to evaluate microbial species and abundance in the intestinal microbes of *T. ovatus*, and the results were aligned against the CARD database. This study revealed the impact of deep-sea culture on aquatic product growth and development, intestinal microbe species and abundance, and antibiotic resistance diffusion patterns, which provides a basis for the scientific development and rational layout of deep-sea aquaculture.

## 2. Materials and Methods

### 2.1. Fish Sampling

Deep-sea cultured fish samples were collected from deep-sea nets more than 50 nautical miles off the coast of Beibu Gulf, Guangxi, China, in water depths of more than 10 m, with strong winds and currents. In contrast, coastal fish samples were collected from traditional net pens located less than one nautical mile off the coast of Beibu Gulf. In late May, fry from two different sampling sites were released and fed the identical diet. In accordance with the growth cycle of *T. ovatus*, the experimental fish were collected after one month, three months and five months of fry placement for subsequent experiments.

### 2.2. Sample Collection

Whole fish: At each sampling, 20 fish were randomly selected from each cage. Samples were transferred to a sterile environment immediately after collection at the sampling site for the next stage of processing.

Tissues and organs: The fish’s liver, kidney, muscle, and intestine were collected immediately into a sterile tube and snap frozen by liquid nitrogen. The samples were exposed to liquid nitrogen for no more than 10 s. For further examination, the samples were kept in the refrigerator at −20 °C [21].

### 2.3. Growth Performance Analysis

In the growth performance test on 20 fish, the following variables were calculated:Body length and weight;Weight gain rate (WGR, %) = 100 * (final body weight − initial body weight)/initial body weightSpecific growth rate (SGR, % day^−1^) = 100 * (Ln final weight − Ln initial weight)/number of days [22];

### 2.4. Analysis of Fish Physiological and Biochemical Indicators

The physiological and biochemical indicators test on 5 fish. 

Liver, kidney, and muscle tissues were added to PBS at pH 7.30–7.50 and then ground into a homogenate. The homogenate was placed in a centrifuge tube and centrifuged at 3500 rpm for 6 min, after which the supernatant was aspirated. Then, the contents of C3, C4, SOD and LYS in fish tissues were determined by a kit provided by ELISA detection system (Jingmei, Biotech, Yancheng, Jiangsu Province, China). Microplate reader provided by public service platform for large-scale instruments and equipment of College of Life Sciences and Oceanography, Shenzhen University.

### 2.5. Analysis of Water Quality

Analytical items including pH, dissolved oxygen (DO), chemical oxygen demand (COD), inorganic nitrogen, and reactive phosphate were studied to evaluate the quality of water. The field processing and analytical measurements of the 10 L water were performed according to the standard methods [23].

### 2.6. Genomics DNA Extraction

The analysis of the intestinal microbes of *T. ovatus* by metagenomics test on 5 fish.

The microbial community DNA was extracted using MagPure Stool DNA KF kit B (Magen, Guangzhou, China) following the manufacturer’s instructions. DNA was quantified with a Qubit Fluorometer by using Qubit dsDNA BR Assay kit (Invitrogen, CHI, Carlsbad, CA, USA) and the quality was checked by running aliquot on 1.00% agarose gel [24].

### 2.7. Library Construction

According to Zhuet al. [25], the main steps of library construction are shown as below: (1) Sample testing. Sample testing includes the concentration, integrity, and purity of the sample. Concentration is determined by fluorescence quantification or enzyme standardization. The integrity and purity of the samples are measured on agarose gels (agarose gel concentration: 1%, voltage: 150 V, electrophoresis time: 40 min). (2) Sample interruption. Take 1 μg of genomic DNA and disrupt it using a Covaris instrument with ultrasound. (3) Fragment size selection. The beads of the interrupted sample are selected to concentrate the sample bands around 200–400 bp. (4) End repair, A base addition and splice ligation. (5) Prepare the reaction system and react for a certain time at the appropriate temperature, repair the end of the double-stranded cDNA and add A base to the 3’ end. Prepare the reaction system, react for a certain period of time at moderate temperature to connect the junction to the DNA. (6) PCR reaction and product recovery. The PCR reaction system is prepared and the reaction program is set up to amplify the ligated products. The amplified products are purified and recovered by magnetic beads. (7) Product cyclization. After denaturing the PCR products to single-stranded, the cyclization reaction system is prepared and mixed thoroughly for a certain period of time at the appropriate temperature to obtain the single-stranded cyclic products, and the final library is obtained by digesting the uncyclized linear DNA molecules. (8) Library detection. The cyclized product is tested for concentration before loading. (9) On-board sequencing. A single-stranded circular DNA molecule is replicated by ring rolling to form a DNA nanoball (DNB) containing multiple copies. The obtained DNBs are added to the mesh pores of the chip using high-density DNA nanochip technology and sequenced by co-probe anchored polymerization (cPAS).

### 2.8. Metagenomic Data Analysis

All the raw data were trimmed by SOAPnuke v.1.5.2 [26]. High-quality reads were de novo assembled using Megahit software. Assembled contigs with length less than 300 bp were discarded in the following analysis [27]. Genes were predicted over contigs by using MetaGeneMarker (2.10) [28]. Redundant genes were removed using CD-HIT [29] with identity cutoff 95%. To generate the taxonomic information, the protein sequences of genes were aligned against the NR database using DIAMOND [30] with an E value cutoff of 1 × 10^−5^. Based on the MEGAN [31] LCA algorithm, the taxonomic annotation was assigned. To obtain functional information, the protein sequences were aligned against the eggNOG database (2015-10), CAZy database (2017-09), COG database (2014-11), Swiss-prot database (2017-07), KEGG database (89.1) and CARD database (4.0) by DIAMOND [5] with an E value cutoff of 1 × 10^−5^. To generate the taxonomic and functional abundance profiles, the reads were aligned to the genes using Botwie 2 [32] with the default setting. Based on the abundance profiles, the features (Genera, Phyla and KOs) with significantly differential abundances across groups were determined using Wilcoxon’s rank sum test [33]. *p*-values for multiple testing were corrected using the BH [34] method with corrected *p*-values < 0.05 considered significant. Differentially enriched KEGG pathways were identified according to the reporter scores [35]. An absolute value of reporter score of 1.65 or higher was used as the detection threshold for significance. The alpha diversity was quantified by the Shannon index using the relative abundance profiles at gene, genus and KO levels with R package. The beta diversity was calculated using Bray–Curtis distance [36] or Jensen–Shannon Divergence distance [37]. 

The above analysis software was provided by Shenzhen University Testing Centre.

### 2.9. Analysis of Experimental Data

To examine the effects, all experimental data are reported as means ± standard deviation (S.D.) and subjected to one-way ANOVA independent-sample f-test and Duncan multiple comparisons using the SPSS for Windows program (ver 16.0, CHI, USA). Unless otherwise stated, statistical significance was determined at a level of *p* < 0.05.

### 2.10. Ethics Statement

The studies involving animals were reviewed and approved by the College of Life Sciences and Oceanography, Shenzhen University. 

## 3. Result

### 3.1. Analysis of Water Quality

The water quality indicators of coastal and deep-sea water are presented in Table 1. The pH of natural seawater is often stable between 7.90–8.40, and unpolluted seawater is between 8.00–8.30. Water quality tests showed that the pH of the water in the deep sea was 8.19, 8.22 and 8.13 in the months of June, August, and October, respectively, all at normal levels. However, in August and October the pH of the water bodies in the coastal area was below 7.90, which was outside the normal range and considered polluted. The content of DO in seawater is one of the important parameters of seawater quality [38]. Furthermore, the DO levels of the water bodies in both areas fluctuated but were basically maintained at normal levels. As an important indicator of water pollution, COD represents the content of organic substances in water; the greater the COD, the more serious the pollution of water bodies by organic substances [39]. The COD content of coastal water bodies reached more than twice that of deep-sea water all the time, and even reached 3.10 times in October, exceeding the normal range. This indicates higher organic matter pollution in coastal water than in deep-sea water. In addition, inorganic nitrogen and reactive phosphate in seawater can be used and decomposed by organic matter, which makes them important indicators to evaluate the degree of seawater eutrophication. The findings revealed that inorganic nitrogen (October, 0.44 mg/L) and reactive phosphate (October, 0.058 mg/L) levels in the coastal area’s water bodies were 110 times and 29 times higher than those in deep-sea water bodies, and the water bodies were highly eutrophic. In summary, over the 150-day feeding cycle, the deep-sea group outperformed the coastal group in all indicators, showing better water quality, especially in August and October, which indicated that the water in the coastal area was highly polluted and not suitable for aquaculture. After measuring the pH, DO, COD, inorganic nitrogen, and reactive phosphate of the seawater in the two areas, it is obvious that the seawater in the coastal area was seriously polluted by microorganisms and organic matter, and showed a eutrophic state [40]. Pesticides, chemical plants, and organic fertilizers are the main sources of organic pollution, and the higher the chemical oxygen demand, the more serious the organic pollution in the water body [41]. At the same time, these organic pollutants can also lead to the deposition of inorganic nitrogen, reactive phosphorus, and other substances in the water body [42], causing eutrophication [43]. If people were fed with the fish raised in polluted water, they would absorb and accumulate in their bodies a lot of toxins from these organisms, which are often carcinogenic, deformogenic, mutagenic, and extremely dangerous to people [44]. It is clear that the levels of COD, inorganic nitrogen, and reactive phosphorus in seawater in August and October in nearshore areas exceeded the upper limits set for aquaculture and were no longer suitable for aquaculture activities. This phenomenon may have been caused by coastal human activities and domestic and industrial water discharges. In summary, the water in the coastal area was environmentally polluted, with a lower pH than normal seawater and higher eutrophication, which leads to a worse habitat for aquaculture in this area.

### 3.2. The Growth, Physiological and Biochemical Characteristics of T. ovatus

The physiological and biochemical indicators of fish reared in different regions are shown in Figure 1 and Figure 2. There is no significant effect of coastal and deep-sea culture on the growth of fish. As shown in Figure 1a, although the coastal group was 13.03% heavier than the deep-sea group at the beginning of the culture period (30 days), the data at days 90 and 150 showed that the weight of *T. ovatus* raised by the two culture methods was close. Similar to body weight, the body length of fish (Figure 1b) from deep-sea culture was 8.43% lower than that of fish from coastal culture at the beginning of growth, but after 150 days of rearing, the body length exceeded that of fish from coastal group (1.19% more). The initial length and weight of the fish raised in coastal waters (19.60 g and 10.80 cm, respectively) were (13.03 and 8.43%, respectively) higher than that of the fish (17.34 g and 9.96 cm, respectively) raised in the deep sea. Although there was no significant difference in the net length and weight between coastal and deep-sea samples on the 150th day, the rates of increase in length and weight (from day 30 to day 150) of deep-sea farming (2768.80% and 198.69%, respectively) were significantly higher than those of coastal fish (2447.96% and 172.13%, respectively). Results indicated that by aquaculture farming in the deep sea, *T. ovatus* can grow faster and more efficiently, compared with coastal farming. Moreover, the WGR and SGR verified the initial results (Figure 1c,d). WGR reflects the relationship between the animal’s weight gain and initial body weight over a period of time, and can visually reflect the animal’s weight gain and growth [45]. SGR is the ratio of growth rate to the number of days of growth, which is a common indicator to measure the growth status. A greater specific growth rate means that a living individual gains more weight per day, effectively reducing the influence of the individual’s initial weight in the calculation of growth performance [22]. Both WGR and SGR were lower (13.70% and 4.3% less, respectively) in the deep-sea group during the early rearing period (from 1 to 30 days); however, after 150 days of rearing, there was no significant difference observed between the coastal and deep-sea groups, indicating that the deep sea is favorable for the fish’s growth rate. The complements C3 and C4 are markers of acute inflammation and tissue damage. LYS is important for non-specific immunity, while SOD measures the body’s antioxidant capability. The results showed that there were no visible trends and differences in the aforesaid physiological indicators (C3, C4) in various tissues of *T. ovatus* from both regions during 150 days of rearing. However, the levels of LYS in all tissues of the deep-sea group were higher than those of the coastal group for most of the time, and the levels of SOD were also equal to or even higher than those of the coastal group. This indicated that the antioxidant capacity and non-specific immunity of *T. ovatus* cultured in coastal waters could be enhanced, possibly due to the exposure of fish to various pollutants and microorganisms. The differences in growth and development of *T. ovatus* reared in different sea areas by measuring the growth performance and feed-utilization efficiency indexes were investigated. The results showed that one month after the fry were released, the growth indicators such as the fish’s bodyweight reared in the deep sea were lower than those reared in coastal waters. *T. ovatus* are expected to spend about a month in coastal high-salinity conditions before undergoing metamorphosis and transition to juveniles [46], but in the deep sea, the intense water-exchange environment and wind and waves could have some stressful effects on the fish, which leads to a more delayed development [47]. The growth limitation was compensated after five months, and there were no significant variations in growth performance or feed-consumption efficiency between the two groups. After reaching a bodyweight of 450 g, the growth efficiency and feed-utilization efficiency of *T. ovatus* decreased dramatically and the mortality rate increased, so feeding is usually ended and the fish are sold as a commodity after 5–6 months of feeding [47]. In summary, although the fish in the deep-sea group were smaller and slightly lower in length and weight than the coastal groups after fry placement and early rearing (within 30 days), these gaps were closed after 90 and 150 days of rearing in the deep sea. As the results showed, SOD and other physiological and biochemical indicators of *T. ovatus* cultured at different times and areas fluctuate within a relatively normal range, and there is no more obvious trend of change. In addition to this, the mortality of fish between the two areas was more stable, and there were no obvious lesions or infections in appearance [48]. However, for most of the time, fish fed in the deep-sea group had higher levels of C3 complement protein and LYS in all parts of the fish than the coastal group, and similar or even higher levels of SOD than the coastal group. To further explore the effects of various aquaculture farming strategies on the physiological and biochemical characteristics of *T. ovatus*, C3, C4, SOD and LYS, which are non-specific immune indices [49], were also analyzed in this article. Results showed that no significant differences were observed for C3, C4, and SOD between coastal and deep-sea fish. However, deep-sea farming showed much better results (increase of 7.07%) for LYS, compared with coastal fish. Since the LYS increased for the fish raised in deep-sea farming, it indicates that the deep-sea cultured fish had better immunity and disease resistance.

### 3.3. Metagenomics Analysis of Intestinal Microbes of T. ovatus

Intestinal microbes are believed to have an important impact on host life activities, and play a prominent part in host immunity, adaptation, and nutrient metabolism [50]. The species abundance of intestinal microbes at various taxonomic levels is shown in Figure 3. Through sequencing, library creation, and screening procedures, a total of 1,937,462/1,820,444 (coastal/deep-sea) microbial sequences were found in the intestinal contents of fish bred in coastal and deep-sea waters, respectively. The sequences comprise 4 kingdoms, 54 phyla, 106 classes, 211 orders, 475 families, 1576 genera, and 5061 species. There were 1,068,238 (55.50%)/1,010,303 (55.14%) eukaryota sequences, 825,677 (42.21%)/768,440 (42.62%) bacterial sequences, 16,555 (0.87%)/15,820 (0.85%) archaeal sequences and 26,992 (1.42%)/25,881 (1.39%) virus sequences obtained. We examined the proportion of microorganisms identified at three taxonomic levels (Figure 3). The prevailing phylum was *Ascomycota* (52.98/53.32%), followed by *Pseudomonadota* (24.30/22.13%), *Bacillota* (10.39/11.82%), *Bacteroidetes* (3.00/3.13%), and *Basidiomycota* (2.60/2.61%). At the family level, 15.10/12.98% of reads were classified as the family *Pasteurellaceae*, 13.31/12.85% of reads were classified as *Glomerellaceae,* and 10.69/10.93% of classified reads were classified as *Saccharomycetaceae*. However, the rest of the families with the largest percentage in the coastal group were *Pyriculariaceae* (6.94%) and *Nectriaceae* (6.64%), and in the deep-sea group were *Bacillaceae* (7.58%) and *Pyriculariaceae* (7.24%). At the genus level, in the coastal group, the genus with the highest percentages was *Mannheimia* (14.81%), followed by *Colletotrichum* (13.24%), *Pyricularia* (6.90%), *Fusarium* (6.60%) and *Bacillus* (6.08%); meanwhile, for the deep-sea group it was *Colletotrichum* (12.78%), *Mannheimia* (12.64%), *Pyricularia* (7.20%), *Bacillus* (7.16%) and *Fusarium* (6.66%). 

### 3.4. Diversity Analysis

The alpha diversity index of the intestinal flora of *T. ovatus* is shown in Table 2. The Chao1 index measures the diversity of a community, with a higher number indicating greater diversity [51]. The Shannon and Simpson Diversity Indexes measure the community’s diversity, with higher values indicating more diversity [52]. Alpha diversity results showed that only the Simpson index for October in the deep-sea group was slightly lower than the other groups, while the Chao1 and Shannon indexes in the other groups were not significantly different [53]. A higher diversity of intestinal microbial species indicates a more complex microbial structure and a more stable homeostasis of the intestinal environment [54]. The results showed that the Chao1 index of microorganisms in the fish gut in both regions did not show a clear pattern, although the temporal variation was relatively slight. However, unlike the coastal group, where the Shannon and Simpson indexes remain relatively steady, the deep-sea group’s indexes indicate a downward tendency. The results of the principal coordinates analysis are shown in Figure 4. The distance between sample points in the graph was used to determine the differences between individuals or groups. The greater the distance between sample points, the greater the difference in microbial community composition between samples, and the closer the distance, the more similar the microbial community composition between samples [55]. The results show that the gut microbes in the deep-sea region are structurally similar and have some stability in the four months. In contrast, the structure of the gut microbial community in the coastal group was more variable and less similar during the four months. In addition, the gut microbial structure of fish in the two regions in June also had similarities. Combining the results of the alpha diversity and beta diversity analyses, it can be seen that the gut microbial diversity and structure in the deep-sea region remained stable during the culture period and did not produce significant changes. However, the gut microbial structure of fish cultured in the coastal region, although maintaining a considerable diversity, has undergone significant changes in its composition. Although many existing studies have shown that a decrease in gut microbial diversity can have a major influence on host health [56], given the small magnitude of the decline in microbial diversity in the deep-sea group and the long period of time before the development of individuals, it cannot be concluded that this trend is necessarily detrimental. 

### 3.5. Antibiotic Resistance Gene Analysis

A total of 24 antibiotic resistance genes (ARGs) were obtained and classified into four genotypes, including three single resistance genes and two multiple resistance genes. Among them, 2 resistance genes were found in the deep-sea group, each of which was a genotype, and 23 resistance genes were found in the coastal group, which could be classified into 4 genotypes (Figure 5) The widespread use of antibiotics for disease prevention and treatment has had a negative impact on the environment, and intestinal microbes. Meanwhile, antibiotic resistance genes are also common in aquaculture, posing a serious threat to aquatic organisms and humans [57]. The resistance gene FabG, in the deep-sea group, is a 3-oxoacyl carrier protein reductase which plays a role in lipid metabolism [58]. Resistance to triclosan can be caused by point mutations in the FabG gene. Ribosomal protein S12 (RpsL) stabilizes the highly conserved pseudoknot structure formed by 16S rRNA [59]. Amino acid substitutions in RpsL affect the higher-order structure of 16S rRNA and create resistance to streptomycin, a class of aminoglycoside basic antibiotics [60]. In addition, we found the gene sequence of acid shock protein Asr in the intestinal flora of the coastal group. Acid shock protein Asr can function in environments with low pH and can enable cells to survive at extremely low pH (pH 2.0) [61]. Cation/multidrug efflux pump AdeG is part of the AdeFGH efflux system [62]. It confers resistance to antibiotics such as tetracyclines, chloramphenicol, fluoroquinolones, trimethoprim, and other antibacterial biocides such as dyes and anionic detergents, and has been widely used in aquaculture [63]. ARGs were found in the intestines of fish from both regions, suggesting that the water bodies in both regions have received different levels of antibiotic contamination. The deep-sea region was less contaminated and had fewer numbers and types of resistance genes. From the above results, it is clear that the species of ARGs in the intestinal flora of *T. ovatus* cultured in coastal areas are higher than those in deep-sea areas. This also suggests that coastal-farmed fish may be enriched with more antibiotics or pathogenic bacteria in their intestines, which not only adversely affects the health of aquatic products, but may also pose a potential threat to the health of human consumers.

## 4. Discussion

### 4.1. Association of Environmental Factors with Intestinal Flora

Peripheral water environmental factors play an important role in shaping the composition and stability of the intestinal microbiome, thereby directly or indirectly influencing host energy metabolism, nutrient absorption, immune responses, and other physiological conditions [64]. Intestinal microorganisms of marine animals have been beneficial to the host’s healthy, physiology and immune response [65]. However, the homeostasis of fish gut microorganisms is fragile, and changes in external or internal factors (gut structure, water temperature, dietary factors) can affect the diversity and structural composition of fish gut microorganisms [66]. The typical microorganisms found within the intestinal flora of fish are bacteria, consisting primarily of *Bacillota*, *Pseudomonadota*, and *Bacteroidetes* [67], aligning with our results. The peripheral environment typically exerts two effects on the intestinal microbial structure of aquatic organisms, either by inhibiting or promoting the growth of microorganisms, or by regulating the metabolism of microorganisms in the habitat [68]. The environmental stresses such as pollution, low oxygen, and sudden temperature changes can compromise the host immune system, leading to pathogen invasion and changes in the composition of gut microbes. For instance, the diversity of microorganisms in the gut of healthy fish was higher than that of unhealthy fish, and the abundance of metazoan phyla in healthy fish was lower than that of diseased fish, with *Aeromonas* possibly being the key cause of the infectious disease [33]. Interestingly, *Ascomycota* is generally found in the gut of freshwater fish and rarely in the gut of marine fish [69]. This is also different from what has been described in prior research on fish gut microbiology [54], although there have also been studies that indicate considerable changes in the gut microbial composition of wild and laboratory-raised fish [70]. Tiunov et al. showed that some fungi of the *Ascomycota* phylum such as *Debaryomycetaceae*, *Candida*, and *Saccharomyces* are important to the host in promoting growth, catabolism, and nutrient redistribution [71]. Furthermore, these fungi play a role in fish host defense. Compared to non-colonized larvae, zebrafish larvae pre-colonized with different yeast strains had a better immune response and survival against the pathogen *Vibrio anguillarum* [72]. As mentioned earlier, the parameters tested in the water column indicated that there was some eutrophication in the water column of the coastal area. And some studies have shown that the *Pseudomonadaceae* and *Aspergillus* families occur in significantly higher proportions in the gut microbiome under eutrophic conditions, which is consistent with the results of this study [73]. The study found that the common lactic acid bacteria in the gut of marine fish and fresh-water fish are usually not the dominant intestinal flora, but after habitat changes (such as when they are cultured in a specific pond culture system), the lactic acid bacteria will become the dominant intestinal flora and remain stable, so as to improve the adaptability to the new habitat and enhance the immune ability. According to species annotation results, the components of the intestinal microbes of *T. ovatus* cultured in the two regions were similar. At the kingdom level, although the total number of microorganisms in the coastal group was more diverse, the proportion of the four communities was similar in both regions. Moreover, at the phylum level, the dominant bacteria in each group were *Ascomycota*, *Pseudomonadota*, *Bacillota* and *Bacteroidota*; among them, *Pseudomonadota*, *Bacillota* and *Bacteroidota* are consistent with previous studies. And at the family level, we also found the aforementioned probiotic bacteria, so we speculate that *Ascomycota* is generally beneficial in the gut of *T. ovatus*. Although some fungi are considered pathogenic, the *T. ovatus* reared in both areas of this experiment were apparently healthy and did not exhibit fungal infection. So, we believe that some pathogenic fungi are symbiotic with probiotic bacteria and have potential pathogenicity, which may infect the host and make it pathogenic if the host is immunocompromised. So, although water quality tests have shown that waters in coastal areas have received eutrophic pollution with higher levels of microorganisms, organic nitrogen, and inorganic phosphorus, this did not have a serious impact on fish growth, immunity and gut microbial structure and species abundance. However, the potential health risks to consumers of these fish products produced in poor environments are a concern, and there is no doubt that fish raised in more environmentally benign distant marine areas are more reassuring [74].

### 4.2. Effect of Different Culture Areas on Antibiotic Resistance Genes in the Intestinal Flora of Fish

In aquaculture environments, ARGs spread with the migration of flora and can enter directly into the cultured organisms and eventually undergo horizontal gene transfer through the food chain and human gut microbiota [75]. After entering the intestinal tract, antibiotics may cause massive death of probiotic bacteria, severely disrupting the homeostasis of the intestinal environment and thus adversely affecting the health of the host [76]. The spread of ARGs in the environment and in living organisms has caused increasing concern in society due to the long-lasting and persistent risk of ecological and food-safety uncertainty [57]. In the present study, we found ARGs in the gut microbes of *T. ovatus* cultured in both regions. Interestingly, the resistance genotype FabG of the antibiotic triclosan, which is widely used in terrestrial agricultural production, was found in the gut of fish from the deep-sea group, but was not identified in the coastal group [77]. This is most likely due to improper handling during feeding, and the presence of triclosan in the environment due to diffusion by ocean currents is less likely. The resistant genotype RpsL was detected in the intestinal microbes of both regions; however, the coastal group had a far larger number than the deep-sea group. This genotype may be due to the presence of streptomycin in the environment, which leads to mutations in the coding sequence of this protein [78]. The genotype of the acid shock protein Asr, which enables cells to survive in a lower pH environment, can be detected most likely due to sewage discharged from coastal industries. Asr grows and induces acid tolerance at moderate acidity (pH 4.5) and acidic industrial wastewater, especially hydrochloric acid, provides the environmental basis for this reaction [52]. In turn, the microorganisms that have developed acid resistance then enter seawater through the discharge of industrial effluents and are then enriched in the fish’s gut [79]. Whereas the tetracycline resistance genotypes AdeF and AdeG detected only in the nearshore group have been widely used in aquaculture, land-based pond aquaculture is the most likely source of this genotype [80]. And a study of the correlation between environmental microbial species and antibiotic resistance genes showed a significant positive correlation between *Pseudomonadota* and a variety of tricyclic, eosinophilic antibiotic resistance genes [81]. This is consistent with the up-regulation of the proportion of *Pseudomonadota* in the gut microbes of fish from nearshore areas. Therefore, we hypothesize that *Pseudomonadota* is likely to be the main source of resistance genes in the fish gut. Although resistance genes were identified in the gut in both regions, they were less diverse and numerous and had no impact on host health or growth. However, this indicates that the water bodies in both regions are contaminated with antibiotics, only to a different extent, and that aquaculture farmed in their regions is at risk of spreading antibiotic resistance genes. And whether bacteria carrying antibiotic resistance that are not pathogenic to fish can cause disease or transfer antibiotic resistance to humans through the food chains also needs further study. In contrast, the lower number of resistance genes in the gut of deep-sea cultured *T. ovatus* suggests less pressure from environmental antibiotic contamination and greater safety for consumption.

## 5. Conclusions

We investigated the effects of deep-sea farming and traditional coastal aquaculture on aquatic products by comparing the growth performance, physiological indicators, differential gene expression, and intestinal flora microbes. The deep-sea culture enhances the fish’s growth and improves the fish’s antioxidant activity and non-specific immunity. Correlation analysis of intestinal flora and environmental factors showed that lower water pollution and eutrophication levels resulted in a higher abundance of species in the gut flora of fish cultured in deep-sea areas, with higher proportions of probiotic bacteria than the coastal fish. ARGs in the intestinal of fish from deep-sea aquaculture were significantly reduced compared to those from coastal aquaculture, suggesting a lower level of antibiotic contamination in the distant marine environment and a higher level of edible safety of the produced fish products. Our results help to understand the relationship between environmental differences in aquaculture in deep-sea areas and the growth immune performance of aquatic products, changes in the structure of the gut microbial community, and the acquisition of antibiotic resistance genes, and provide a reference for a more scientific and environmentally friendly development of aquaculture.

## Figures and Tables

**Figure 1 animals-13-03365-f001:**
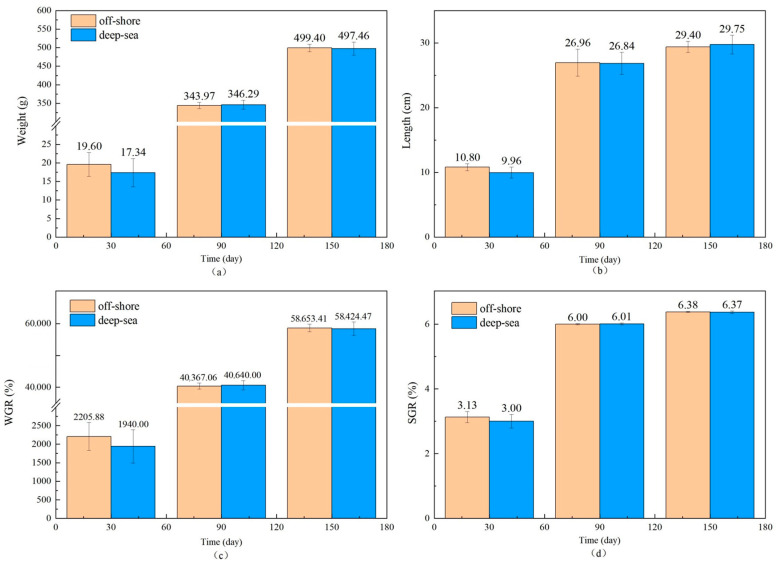
The growth performance of *T. ovatus* ((**a**–**d**) are the results of changes in fish weight, body length, WGR and SGR over time, respectively. WGR for weight gain rate, SGR for specific growth rate. Error bars represent the standard error of the mean).

**Figure 2 animals-13-03365-f002:**
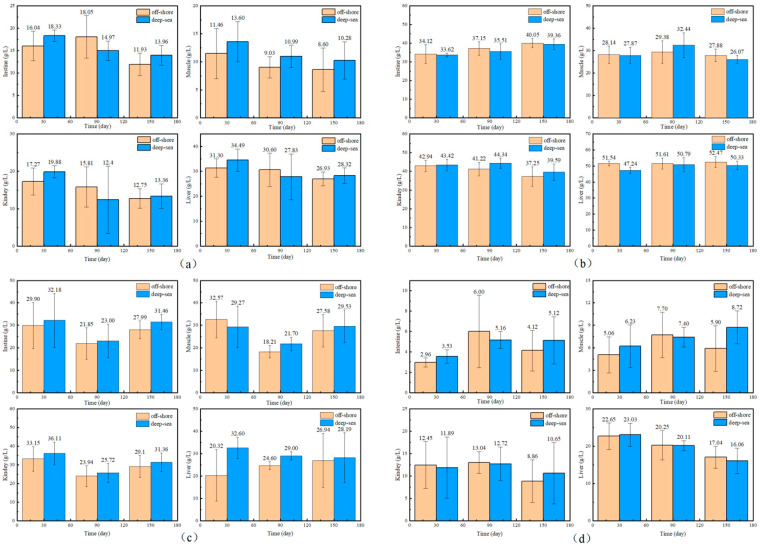
The biochemical indicators of *T. ovatus* ((**a**) for C3, (**b**) for C4, (**c**) for LYS, (**d**) for SOD, in each illustration, from left to right and from top to bottom, are the levels of each indicator in the intestine, muscle, kidney and liver of the fish. Error bars represent the standard error of the mean).

**Figure 3 animals-13-03365-f003:**
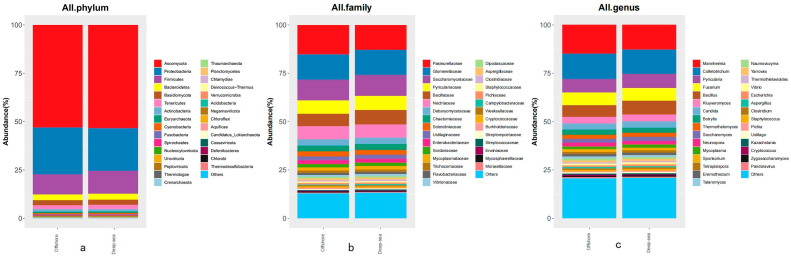
The species abundance of intestinal microbes at various taxonomic levels ((**a**–**c**) are the top thirty species with the highest species abundance at different taxonomic levels in the gut microbes of fish in each of the two regions, with the rest classified as “Others”, (**a**) for phylum level, (**b**) for family level and (**c**) for genus level).

**Figure 4 animals-13-03365-f004:**
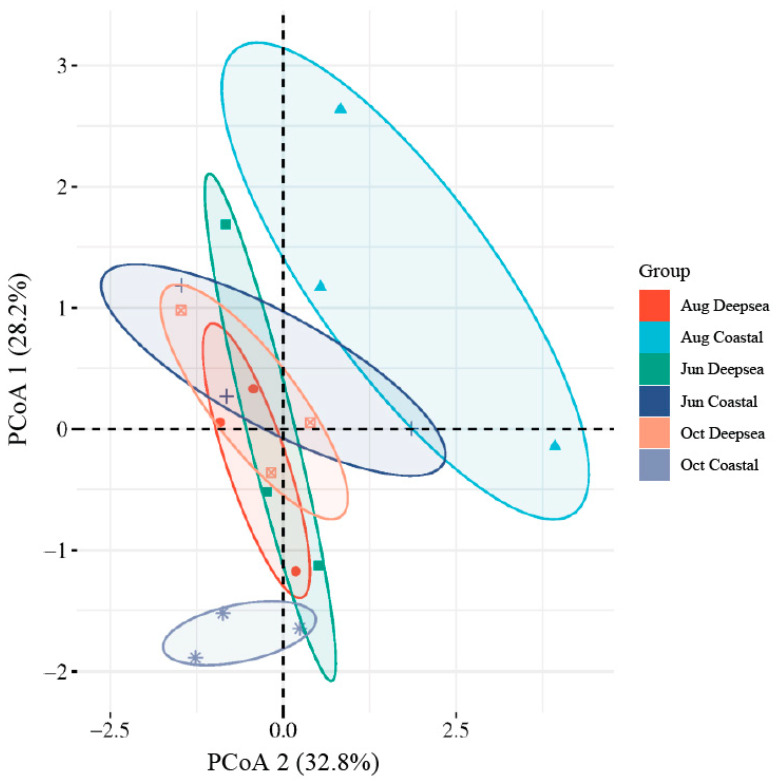
The Principal co-ordinates analysis (PCoA) plot of intestinal microbials. Confidence ellipses were separately constructed for the gut microbiota at both sampling sites and different time periods (*p* value < 0.01). An increased degree of overlap in the confidence ellipses suggests a closer alignment in the gut microflora structure between the groups.

**Figure 5 animals-13-03365-f005:**
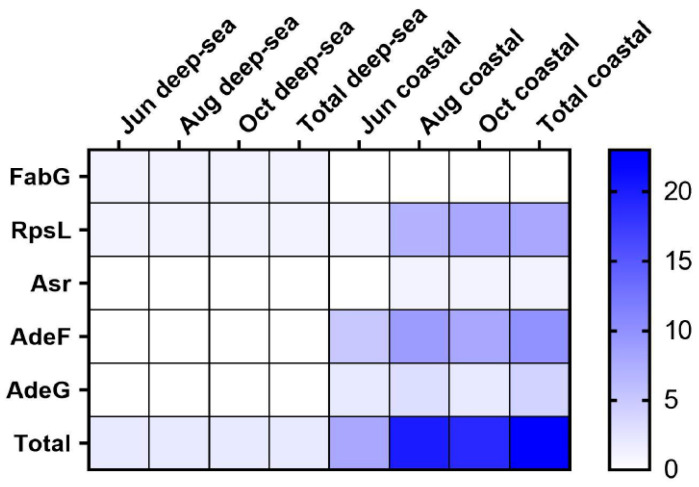
The heat map of antibiotic resistance genes in the intestinal microbial community (FabG for 3-oxoacyl carrier protein reductase, Rpsl for Ribosomal protein S12, Asr for acid shock protein, and AdeF and AdeG for Cation/multidrug efflux pump).

**Table 1 animals-13-03365-t001:** The water quality indicators of different regions (DO for dissolved oxygen, COD for chemical oxygen demand).

Group	Time	pH	DO (mg/L)	COD (mg/L)	Inorganic Nitrogen (mg/L)	Reactive Phosphate (mg/L)
Coastal	June	8.02	7.95	1.27	0.18	0.008
	August	7.86	7.46	1.50	0.32	0.032
	October	7.70	6.41	1.24	0.44	0.058
Deep-sea	June	8.19	6.96	0.50	0.045	0.003
	August	8.22	6.60	0.54	0.006	0.003
	October	8.13	7.46	0.40	0.004	0.002

**Table 2 animals-13-03365-t002:** Intestinal microbial Alpha diversity index in *T. ovatus*.

Diversity Index	Group					
	Jun Deep-Sea	Aug Deep-Sea	Oct Deep-Sea	Jun Coastal	Aug Coastal	Oct Coastal
Chao1_index	42	40	44	42	43	41
Shannon_index	1.22	1.19	1.13	1.16	1.2	1.19
Simpson_index	0.57	0.58	0.55	0.58	0.58	0.58

## Data Availability

The data presented in this study are available in article.

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
