# Peer review of "Differences in Physiological Performance and Gut Microbiota between Deep-Sea and Coastal Aquaculture of Thachinotus Ovatus: A Metagenomic Approach"

_animals, 2023, doi:10.3390/ani13213365_

Round 1

Reviewer 1 Report

Simple summary is instructions, not the paper summary

“Nearshore” and “off-shore” seem to be used interchangeably, yet ‘off shore’ often means ‘far from shore’, the area that the authors term “deep sea”. Terms should be used consistently and defined by water depth and distance from shore. Perhaps “coastal” would be a better contrasting term than “off-shore” with “deep sea.”

Figure 2 needs a more detailed caption explaining a, b, c, and d groupings.

More info is needed about the potential environmental impact of deep sea pens on their surroundings, and the concern about release of farmed fish into the wild.

Intro needs some English revision but does explain the purpose of the study. In general the English is ok but some parts of the introduction are difficult to follow. 

Author Response

“Nearshore” and “off-shore” seem to be used interchangeably, yet ‘off shore’ often means ‘far from shore’, the area that the authors term “deep sea”. Terms should be used consistently and defined by water depth and distance from shore. Perhaps “coastal” would be a better contrasting term than “off-shore” with “deep sea.”

Thank you for your valuable input, there are currently different designations for farming methods for different distances from shore and water depths. For example, the CHU(2020) study indicated that farming practices 500-3000 metres offshore with a depth of less than 50 metres should be referred to as off-coast farming, while farming 3000 metres offshore with a depth of more than 50 metres is referred to as off-shore farming. So, we decided to go with your comment to correct off-shore to coastal farming.

Chu, Y.I.; Wang, C.M.; Park, J.C.; Lader, P.F. Review of cage and containment tank designs for offshore fish farming. Aquaculture 2020, 519, 734928, doi:https://doi.org/10.1016/j.aquaculture.2020.734928.

Figure 2 needs a more detailed caption explaining a, b, c, and d groupings.

Thank you for your correction, we have standardised the names of all charts and tables!

Line239: Table 1. The water quality indicators of different regions. Change to Table 1.  The water quality indicators of different regions (DO for dissolved oxygen, COD for chemical oxygen demand)

Line 312: Figure 1. The growth performance of Trachinotus ovatus sp change to Figure 1. The growth performance of Trachinotus ovatus sp. (a, b, c and d are the results of changes in fish weight, body length, WGR and SGR over time, respectively. WGR for weight gain rate, SGR for specifific growth rate Error bars represent the standard error of the mean.)

Line 316: Figure 2. The biochemical  indicators in various tissues of Trachinotus ovatus. change to The biochemical indicators of Trachinotus ovatus sp. (a for C3, b for C4, c for LYS, d for SOD, in each illustration, from left to right and from top to bottom, are the levels of each indicator in the intestine, muscle, kidney and liver of the fish. Error bars represent the standard error of the mean. ).

Line 346: Figure 3. The species abundance of intestinal microbes at various taxonomic levels change to Figure 3. The species abundance of intestinal microbes at various taxonomic levels (a,b and c are the top thirty species with the highest species abundance at different taxonomic levels in the gut microbes of fish in each of the two regions, with the rest classified as “Others”. a for phylum level, b for family level and c for genus level.)

Line 386: Figure 4. The heat map of antibiotic resistance genes in the intestinal microbial community change to Figure 4. The heat map of antibiotic resistance genes in the intestinal microbial community (FabG for 3-oxoacyl carrier protein reductase, Rpsl for Ribosomal protein S12, Asr for acid shock protein, and AdeF and AdeG for Cation/multidrug efflux pump)

More info is needed about the potential environmental impact of deep sea pens on their surroundings, and the concern about release of farmed fish into the wild.

Thank you for your suggestion, we have made corrections to your question in the text, for example:

Line 106-110: Deep-sea farmed fish samples were collected from deep-water nets more than 50 nautical miles off-shore in Beibu Gulf, Guangxi Province, China. The coastal fish samples were collected in traditional nets less than one nautical mile off the coast of Beibu Gulf. change to Deep-sea cultured fish samples were collected from deep-sea nets more than 50 nautical miles off the coast of Beibu Gulf, Guangxi, China, in water depths of more than 10 metres, with strong winds and currents. In contrast, coastal fish samples were collected from traditional net pens located less than one nautical mile off the coast of Beibu Gulf.

Reviewer 2 Report

Title of the manuscript: Differences in physiological performance and gut microbiota between deep-sea and off-shore aquaculture of Thachinotus ovatus: a metagenomic approach.

The manuscript needs to be corrected as there are some silly mistakes regarding sentence formation and spelling errors and also the scientific names needs consideration. There are some queries the author must comply.

1.     Simple summary needs to be added and the section is not written at all.

2.     Abstract: The sentence “How traditional off-shore……………. flora is unclear” needs to be rewritten.

3.     Abstract: “Intestinal flora microbial of deep sea” this sentence needs to be corrected.

4.     Abstract: The scientific names in abstract and throughout the manuscript needs to be italicized.

5.     Abstract: The author written “resistance genes were screened” but resistance to what they did not mentioned.

6.     Materials and Methods: the section 2.1 needs to be improved as there is no information regarding rearing environment and rearing area.

7.     Sample collection: Section 2.2 the temperature of the refrigerator must check.

8.     Section2.6: There is no information on the number of sample collection and number of samples.

9.     Section 2.7: there must be space in the citation.

10.  Section 2.7: the sentence “Take 1ug of genomic DNA…….. instruments” needs to be corrected.

11.  Section 3.1: the words need to be corrected like deep-sea, deep sea or deap sea…., SGR, SRG…. Please correct the words.

12.  Figure1: The caption needs to be improved as it is too poor for a journal.

13.   Figure2: the figure caption is poorly formed.

14.  Section 3.3: Please change the subtitle appropriately.

15.  Section 3.3: what is macrogenomic library creation please describe.

16.  Figure 3: The figure legend is incomplete.

17.  Section 3.5: in the material and methods there is not a single mention about this please explain why and include in the sections.

18.  Section 3.5: please be consistent with the gene names in figure and text.

19.  Section 3.5: please check the spelling of also.

20.  Section 3.5: the author has only predicted the gene function from the metagenomic analysis but there is no validation of the genes by doing qPCR, therefore I believe it should be justified from the author.

21.  Section 4.1: the sentence “The dominant…………… core microorganisms” please check the sentence formation as it does not make any sense.

22.  All the scientific names need to be italicized.

23.  Section 4.1: the author claimed that the habitat change can enhance the immune ability but there is no clear answer to it.

Finally I recommend for a major revision as the manuscript needs to upgrade at the metagenomics and antibacterial resistance gene analysis. The metagenomic approach and the representation is too basic and needs improvement as author must explain each of the parameter.

The english language editing needs to be carried along with sentence formation and common spelling errors.

Author Response

  1.  Simple summary needs to be added and the section is not written at all.   

Thank you for the correction, we have corrected the text where it corresponds.

Line16-29:It is vitally important that scientists are able to describe their work simply and concisely to the public, especially in an open-access on-line journal. The simple summary consists of no more than 200 words in one paragraph and contains a clear statement of the problem addressed, the aims and objectives, pertinent results, conclusions from the study and how they will be valuable to society. This should be written for a lay audience, i.e., no technical terms without explanations. No references are cited and no abbreviations. Submissions without a simple summary will be returned directly. Example could be found at https://www.mdpi.com/2076-2615/6/6/40/htm. change to In this manuscript, we determined the growth performance and physiological and biochemical indices of the aquaculture fish Thachinotus Ovatus reared in deep-sea and coastal, and analysed the species abundance and diversity of the fish intestinal microbes using high-throughput sequencing to screen for antibiotic resistance genes. We also analysed environmental water samples to identify possible reasons for the observed differences between the two environments. The study found no significant difference in growth performance between deep-sea cultured fish and traditional coastal cultured fish during the feeding cycle. However, the physiological and biochemical indices and the number of antibiotic resistance genes in the gut microbial community of deep-sea cultured fish were superior to those of their coastal counterparts. Coastal cultured fish are particularly vulnerable to the impact of domestic water discharged from human activities on shore. This may be the cause of the observed phenomenon. These studies can offer additional data for comparing deep-sea aquaculture with traditional coastal aquaculture and provide a reference for the future development of fishery transformation and scientific standardisation of deep-sea aquaculture.

In this manuscript, we determined the growth performance and physiological and biochemical indices of the aquaculture fish Thachinotus Ovatus reared in deep-sea and coastal, and analysed the species abundance and diversity of the fish intestinal microbes using high-throughput sequencing to screen for antibiotic resistance genes. We also analysed environmental water samples to identify possible reasons for the observed differences between the two environments. The study found no significant difference in growth performance between deep-sea cultured fish and traditional coastal cultured fish during the feeding cycle. However, the physiological and biochemical indices and the number of antibiotic resistance genes in the gut microbial community of deep-sea cultured fish were superior to those of their coastal counterparts. Coastal cultured fish are particularly vulnerable to the impact of domestic water discharged from human activities on shore. This may be the cause of the observed phenomenon. These studies can offer additional data for comparing deep-sea aquaculture with traditional coastal aquaculture and provide a reference for the future development of fishery transformation and scientific standardisation of deep-sea aquaculture.

  1. Abstract: The sentence “How traditional off-shore……………. flora is unclear” needs to be rewritten.

Thank you for the correction, we have corrected the text where it corresponds.

Line: 32-34: How traditional coastal and emerging deep-sea farming environments influence aquatic growth, immunity, and intestinal microbial flora is unclear. change to The influence that traditional coastal and emerging deep-sea farming environments exert on aquatic growth, immunity and gut microbial flora is unclear.

  1. Abstract: “Intestinal flora microbial of deep sea” this sentence needs to be corrected.

Thank you for the correction, we have corrected the text where it corresponds.

Line 34-37: To address this question, we compared the growth performance, physiological indicators, differential gene expression, and intestinal flora microbial of deep-sea and coastal aquaculture in Guangxi Beibu Gulf of China. change to To address this question, we compared the growth performance, physiological indicators and intestinal microbiological differences of deep-sea and coastal aquaculture in Guangxi Beibu Gulf of China.

  1. Abstract: The scientific names in abstract and throughout the manuscript needs to be italicized.

Thank you for the correction, probably due to typographical reasons, there was a problem with the formatting of all the scientific names in the manuscript, and we have corrected them all.

  1. Abstract: The author written “resistance genes were screened” but resistance to what they did not mentioned.

Thank you for the correction, we have corrected the text where it corresponds.

Line 42-43: Aligned against the CARD database, a total of 23/2 (coastal/deep-sea) resistance genes were screened and grouped into 4/2 genotypes. change to Aligned against the CARD database, a total of 23/2 (coastal/deep-sea) antibiotic resistance genes were screened and grouped into 4/2 genotypes.

  1. Materials and Methods: the section 2.1 needs to be improved as there is no information regarding rearing environment and rearing area.

Thank you for the correction, we have corrected the text where it corresponds.

Line 106-110: Deep-sea farmed fish samples were collected from deep-water nets more than 50 nautical miles off-shore in Beibu Gulf, Guangxi Province, China. The coastal fish samples were collected in traditional nets less than one nautical mile off the coast of Beibu Gulf. change to Deep-sea cultured fish samples were collected from deep-sea nets more than 50 nautical miles off the coast of Beibu Gulf, Guangxi, China, in water depths of more than 10 metres, with strong winds and currents. In contrast, coastal fish samples were collected from traditional net pens located less than one nautical mile off the coast of Beibu Gulf.

  1. Sample collection: Section 2.2 the temperature of the refrigerator must check.

Thank you for the correction, we have corrected the text where it corresponds.

Line 120-121: The samples are exposed to liquid nitrogen for no more than 10 seconds. For further examination, the samples were kept in the refrigerator at -20 oC[21]. change to The samples are exposed to liquid nitrogen for no more than 10 seconds. For further examination, the samples were kept in the refrigerator at -20 oC[21].

  1. 6: There is no information on the number of sample collection and number of samples.

Thank you for the correction, we have corrected the text where it corresponds.

Line 131: The analysis of the intestinal microbes of Trachinotus ovatus by metagenomics test on 5 fish.

  1. Section 2.7: there must be space in the citation.

Thank you for the correction, we have corrected the text where it corresponds.

Line 151-154: According to Zhuet al.[25], the main steps of library construction are shown as below: 1) Sample testing. Sample testing includes the concentration, integrity and purity of the sample. change to According to Zhuet al. [25], the main steps of library construction are shown as below: 1) Sample testing. Sample testing includes the concentration, integrity and purity of the sample.

  1. Section 2.7: the sentence “Take 1ug of genomic DNA…….. instruments” needs to be corrected.

Thank you for the correction, we have corrected the text where it corresponds.

Line156-157: Take 1 μg of genomic DNA and interrupt it by ultrasound using a Covaris instrument. change to Take 1 μg of genomic DNA and disrupt it using a Covaris instrument with ultrasound.

  1. Section 3.1: the words need to be corrected like deep-sea, deep sea or deap sea…., SGR, SRG…. Please correct the words.

Thank you for the correction, we have corrected the text where it corresponds.

Line 209-211: Water quality tests showed that the pH of the water in the deep sea was 8.19, 8.22 and 8.13 in the months of June, August and October respectively, all at normal levels. change to Water quality tests showed that the pH of the water in the deep-sea was 8.19, 8.22 and 8.13 in the months of June, August and October respectively, all at normal levels.

Line 252-253: There is no significant effect of coastal and deep sea culture on the growth of fish. change to There is no significant effect of coastal and deep-sea culture on the growth of fish.

Line 265-267: Results indicated that by aquaculture farming in the deep sea, T. ovatus can grow faster and more efficently, compared with coastal farming. change to Results indicated that by aquaculture farming in the deep-sea, T. ovatus can grow faster and more efficently, compared with coastal farming.

Line 274-278: Both WGR and SRG were lower (13.70% and 4.3% less, respectively) in the deep-sea group during the early rearing period (from 1 to 30 days), however after 150 days of rearing, there was no significant difference observed between the coastal and deep-sea group, indicating that deep-sea is favorable for the fish growth rate. change to Both WGR and SGR were lower (13.70% and 4.3% less, respectively) in the deep-sea group during the early rearing period (from 1 to 30 days), however after 150 days of rearing, there was no significant difference observed between the coastal and deep-sea group, indicating that deep-sea is favorable for the fish growth rate.

  1. Figure1: The caption needs to be improved as it is too poor for a journal.

Thank you for the correction, we have made the corresponding changes to the figure to make it more standardised.

  1.  Figure2: the figure caption is poorly formed.

Thank you for the correction, we have made the corresponding changes to the figure to make it more standardised.

  1. Section 3.3: Please change the subtitle appropriately.

Thank you for the correction, we have corrected the text where it corresponds.

Line 328: 3.3. Analysis of intestinal microbes of Trachinotus ovatus by metagenomics. change to 3.3. Metagenomics analysis of intestinal microbes of Trachinotus ovatus

  1. Section 3.3: what is macrogenomic library creation please describe.

Thank you for the correction, we have corrected the text where it corresponds.

Line 332-334: Through sequencing, library creation, and screening procedures, a total of 1,937,462/1,820,444 (coastal/deep-sea) microbial sequences were found in the intestinal contents of fish bred in coastal and deep-sea, respectively. change to Through sequencing, macrogenomic library creation, and screening procedures, a total of 1,937,462/1,820,444 (coastal/deep-sea) microbial sequences were found in the intestinal contents of fish bred in coastal and deep-sea, respectively.

  1. Figure 3: The figure legend is incomplete.

Thank you for the correction, we have made the corresponding changes to the figure to make it more standardised.

  1. Section 3.5: in the material and methods there is not a single mention about this please explain why and include in the sections.

Thank you for the correction, we have corrected the text where it corresponds.

  1. Section 3.5: please be consistent with the gene names in figure and text.

Thank you for the correction, we have made the corresponding changes to the figure to make it more standardised.

  1. Section 3.5: please check the spelling of also.

Thank you for the correction, we have corrected the text where it corresponds.

  1. Section 3.5: the author has only predicted the gene function from the metagenomic analysis but there is no validation of the genes by doing qPCR, therefore I believe it should be justified from the author.

Thank you for the correction, in fact, we performed qPCR on the samples to validate the data analysed in the genome. The results are consistent with those described in the manuscript, which we have uploaded as an attachment.

  1. Section 4.1: the sentence “The dominant…………… core microorganisms” please check the sentence formation as it does not make any sense.

Thank you for the correction, we have corrected the text where it corresponds.

Line 435-439: The dominant intestinal flora of fish intestines are usually bacteria, and there are usually three core microorganisms: Bacillota, Pseudomonadota and Bacteroidetes[67], which is in the line with our report. In general, the peripheral environment affects the gut microbial structure of aquatic organisms in two ways, either by inhibiting or promoting the growth of microorganisms, or by regulating the metabolism of microorganisms in the habitat[68]. change to The typical microorganisms found within the intestinal flora of fish are bacteria, consisting primarily of Bacillota, Pseudomonadota, and Bacteroidetes[67], aligning with our results. The peripheral environment typically exerts two effects on the intestinal microbial structure of aquatic organisms, either by inhibiting or promoting the growth of microorganisms, or by regulating the metabolism of microorganisms in the habitat[68].

  1. All the scientific names need to be italicized.

Thanks again for the correction, I am sure that all scientific names have been italicised in the initial upload of the manuscript, and we have hereby corrected all minor errors that appeared in the text.

Round 2

Reviewer 2 Report

Dear Authors,

I would finding the manuscript well aligned and nicely corrected as per the suggestions. I would like to recommend it for publication.